# Energy consumption prediction using the GRU-MMattention-LightGBM model with features of Prophet decomposition

**Shaokun Liang**, **Tao Deng, Anna Huang**, **Ningxian Liu, Xuchu Jiang***

School of Statistics and Mathematics, Zhongnan University of Economics and Law, Wuhan, China

* xuchujiang@zuel.edu.cn

## Abstract

The prediction of energy consumption is of great significance to the stability of the regional energy supply. In previous research on energy consumption forecasting, researchers have constantly proposed improved neural network prediction models or improved machine learning models to predict time series data. Combining the well-performing machine learning model and neural network model in energy consumption prediction, we propose a hybrid model architecture of GRU-MMattention-LightGBM with feature selection based on Prophet decomposition. During the prediction process, first, the prophet features are extracted from the original time series. We select the best LightGBM model in the training set and save the best parameters. Then, the Prophet feature is input to GRU-MMattention for training. Finally, MLP is used to learn the final prediction weight between LightGBM and GRU-MMattention. After the prediction weights are learned, the final prediction result is determined. The innovation of this paper lies in that we propose a structure to learn the internal correlation between features based on Prophet feature extraction combined with the gating and attention mechanism. The structure also has the characteristics of a strong anti-noise ability of the LightGBM method, which can reduce the impact of the energy consumption mutation point on the overall prediction effect of the model. In addition, we propose a simple method to select the hyperparameters of the time window length using ACF and PACF diagrams. The MAPE of the GRU-MMattention-LightGBM model is 1.69%, and the relative error is 8.66% less than that of the GRU structure and 2.02% less than that of the LightGBM prediction. Compared with a single method, the prediction accuracy and stability of this hybrid architecture are significantly improved.

## 1. Introduction

Short-term prediction of energy consumption predicts, estimates, analyzes, judges and speculates on the future development of the energy system, mainly by constructing a mathematical model reflecting the internal activities and external connections of the energy system. The prediction and accurate control of energy consumption have become important for energy savings and emission reduction. The prediction of short-term energy consumption data can

**Data Availability Statement:** All relevant data are within the manuscript and its Supporting Information files. We selected the energy consumption data of the US PJM regional energy

supply company in 14 regions from the Kaggle data website for research (https://www.kaggle.com/datasets/robikscube/hourly-energy-consumption). The energy consumption data from 0:00 on January 1, 2015, to 23:00 on August 3, 2018, were selected, with a frequency of 1 hour, and a total of 31440 samples.

**Funding:** The funders had no role in study design, data collection and analysis, decision to publish, or preparation of the manuscript.

**Competing interests:** The authors have declared that no competing interests exist.

ensure the stable operation of the system and assist the scheduling of energy consumption plans. Therefore, the construction of the energy prediction model is a comprehensive modeling project.

The energy consumption forecasting problem can be regarded as a traditional time series forecasting problem. Luthuli et al. [1] compared conventional energy prediction models and proposed time series decomposition with ANN feature extraction; on this basis, they used the machine learning model SVM method to predict energy consumption. However, the Residual item is discarded, and only Trend and Seasonality were predicted. Its prediction information is not fully exploited. Jain et al. [2] proposed an SVM model based on time window features to predict energy consumption. However, its prediction results are not good enough to give the corresponding evaluation. To capture the nonlinear relationship of time series, Chen, YB et al. [3] proposed support vector regression (SVR) and extreme learning machines (ELMs) to predict energy consumption. Sungwoo Park et al. [4] tried LightGBM in energy prediction based on sliding time window features, which exploited the advantages of ensemble learning and achieved good prediction results.

In addition, the neural network model also has good nonlinear prediction performance. Neural network models can also flexibly adjust feature construction for multistep prediction. Pei, SQ et al. [5] used long-term short memory (LSTM) for multistep forecasting of energy load. Sehovac, L et al. [6] simplified the LSTM model architecture and used a more concise GRU structure, which also achieved good results in energy consumption prediction. Jarábek, T et al. [7] used a recurrent neural network (RNN)-containing decoder to further improve the model performance. Atef, S et al. [8] added a bidirectional structure to the gating mechanism, and the bidirectional LSTM makes the energy consumption prediction more accurate. However, RNN-based models are sensitive to different random weight initializations and thus prone to overfitting. Therefore, the single network structure lacks predictive stability and generalization ability.

Although these single-output forecasting models have demonstrated good forecasting accuracy, they are limited in handling uncertainties such as sudden load changes. To solve the above problems, many studies have begun to combine traditional time models, machine learning models, and neural network models. Park et al. [9] proposed a two-stage STLF model. The first stage included extreme gradient propulsion (XGB) and random forest (RF) prediction methods, and the second stage combined their predictions using multiple linear regression (MLR) models based on sliding Windows. Without sufficient subsampling, randomized trees can lead to overfitting [10]. Xie, Y et al. [11] proposed a two-stage prediction model combining ATT-BiLSTM and MLP. First, a 24-hour prediction was made based on ATT-BiLSTM, and then the results were input into MLP as new features to obtain the final prediction results. However, their deep neural network model is also sensitive to different random weight initializations, which is very unstable and does not combine with machine learning methods. Yuxuan L et al. [12] proposed a method based on the combination of pretrained GRU and LightGBM. However, GRU is used for feature extraction in their work, the periodicity of energy consumption data is not well captured, and GRU-attention has higher prediction accuracy than GRU. Jung S et al. [13] constructed a hybrid model of empirical mode decomposition (EMD) and gated cyclic units (GRUs) based on an attention mechanism. Bu S J et al. [14] proposed a neural network model of an attention mechanism to predict energy consumption. This method only uses an attention structure and has a poor ability to learn the periodic characteristics of power consumption, and the accuracy is not as good as that of GRU-attention.

LightGBM, a single machine learning model, has strong predictive ability in general time series, but its accuracy in predicting periodic time series is still inadequate. Deep learning methods, such as LSTM, GRU and other RNN-based models, rely heavily on initialization

parameter selection, and the model stability and accuracy are not good enough. A single GRU or a single attention is not as good at predicting periodic time series as a combination of the two. Current research cannot solve the above three key problems simultaneously. Based on simple time series decomposition, we propose a feature selection method based on prophet decomposition. The gating-attention mechanism can learn the structure of internal correlation between features. The ensemble learning structure has a strong anti-noise ability and can reduce the impact of mutation points on the overall prediction effect of the model. We proposed a hybrid model architecture of GRU-MMattention-LightGBM. This architecture inherits the advantages of every single architecture and has a more stable prediction effect than a single model.

## 2. Model

### 2.1 Prophet decomposition

Prophet was developed by Facebook's data science team in 2017 (Taylor S J et al. [15]). It used a decomposable time series model (Chung et al. [16]), with three main components: trend, seasonality, and residuals. Although the influences on time series are complex, all series fluctuations can be decomposed into four parts: trend factors $T_t$, cyclic fluctuations $C_t$, seasonal changes $S_t$, and residuals $R_t$.

AC Harveyd et al. [17] simplified the above four items into three items ($T_t$, $S_t$, $R_t$). In this paper, we combined the characteristics of energy consumption data and made the following simplifications in the time series decomposition model, as given by Eq (1).

$$x_t = T_t + S_t + R_t \tag{1}$$

In the Prophet model, the trend term $T_t$ contains two items, namely, the segmental model based on linear regression and the saturated growth model based on logistic regression. The linear part is shown in Eq (2).

$$T_t = (k + \alpha(t)^T \delta)t + (m + \alpha(t)^T \gamma) \tag{2}$$

where $k$ represents the growth rate of the model, $\delta$ is the change in k, $m$ is the offset parameter, $t$ is the timestamp, $\alpha(t)$ is the indicator function, $\alpha(t)^T$ is the transpose vector of $\alpha(t)$, $\gamma$ is the offset of the smoothing process, and its function is to make the piecewise function continuous.

The expression of the saturated growth model based on logistic regression is Eq (3).

$$T_t = \frac{C}{1 + exp(-k(t-m))} \tag{3}$$

where $C$ represents the model bearing capacity, $k$ represents the growth rate, and $m$ is the offset parameter. When the rate $k$ is adjusted, the offset parameter must also be adjusted. Next, the piecewise logistic growth model is Eq (4).

$$T_t = \frac{C(t)}{1 + exp(-(k + \alpha(t)^T \delta)(t - (m + \alpha(t)^T \gamma)))} \tag{4}$$

The Prophet model can incorporate trend changes into the model by setting change points to change the growth rate. Assuming that for timestamp $t$, there are $n$ change points, then $(t) = (\alpha_1(t), \ldots, \alpha_n(t))^T, \gamma = (\gamma_1, \ldots, \gamma_n)^T$. Additionally, for a moment $s_j$, its offset $\gamma_j$, $\gamma_j$ is set to $-s_j \delta_j$. The trend generation model is that there are S rate points in the history of point T, and the rate change of each changing point is $\delta_j \sim Laplace(0, \tau)$.

The seasonal and residual terms can be represented as Eq (5) and Eq (6)

$$S_t = \sum_{n=1}^{N} \left( a_n \, cos\left(\frac{2\pi nt}{P}\right) + b_n \, sin\left(\frac{2\pi nt}{P}\right) \right) \tag{5}$$

$$R_t = x_t - T_t - S_t \tag{6}$$

Zarnowitz V et al. [18] pointed out that *trend, season, and residual* are interrelated and not independent among each other and established a model of interdependence among the three. Therefore, we propose the following more general assumptions, as Eq (7), Eq (8) and Eq (9) are given.

$$T_t = f_1(T_{t-k} \ldots T_{t-1}, S_{t-k} \ldots S_{t-1}, R_{t-k} \ldots R_{t-1}) \tag{7}$$

$$S_t = f_2(T_{t-k} \ldots T_{t-1}, S_{t-k} \ldots S_{t-1}, R_{t-k} \ldots R_{t-1}) \tag{8}$$

$$R_t = f_3(T_{t-k} \ldots T_{t-1}, S_{t-k} \ldots S_{t-1}, R_{t-k} \ldots R_{t-1}) \tag{9}$$

$f_1, f_2, f_3$ are different nonlinear functions, expressed here in the form of an implicit function. Therefore, $T_t, S_t, I_t$, which is determined by the joint action of $T_{t-i}, S_{t-i}, R_{t-i}$, $i \in [1, k]$ *and* $i \in N$. Therefore, the structure of the feature based on Prophet decomposition is Eq (10).

$$F = [T_{t-k} : T_{t-1}, S_{t-k} : S_{t-1}, R_{t-k} : R_{t-1}] \tag{10}$$

$F$ contains *Trend,Season,Innovation* in the first $K$ periods of the energy consumption sequence. Together, they serve as feature inputs to the predictive model. Finally, learn that $f_1$, $f_2, f_3$ are learned and $x_t$ is predicted through the following hybrid architecture.

## 2.2 LightGBM

LightGBM is a variant of the gradient boosting decision tree (GBDT) [19]. We use the Prophet method to extract features and transform the time series forecasting problem of electricity consumption into a supervised learning problem. It is hoped that the learner of ensemble learning can better learn the nonlinear interaction between *the trend, season, and residual* based on the features produced by Prophet. LightGBM is based on the additive model of the boosting strategy. During training, the forward stagewise algorithm is used for greedy learning. Each iteration learns a CART tree to fit the residual between the prediction result of the previous $t-1$ tree and the true value of the training sample. In each combination, the weak learner was better than the previous group. Similar to XGBoost, LightGBM is explicitly regularized. The first half is the loss function, and the second half is the regular term $L_1+L_2$. The approximate objective function is obtained by a second-order Taylor expansion of the loss function, as shown by Eq (11).

$$Obj^t \approx \sum_{i=1}^{N} \left[ l(y_i, \hat{y}_i^{t-1}) + g_i f_t(x_i) + \frac{1}{2} h_i f_t^2(x_i) \right] + \Omega(f_t) \tag{11}$$

Among them $g_i = \partial_{\hat{y}_i^{t-1}} l(y_i, \hat{y}_i^{t-1}), h_i = \partial_{\hat{y}_i^{t-1}}^2 l(y_i, \hat{y}_i^{t-1})$

The GBDT needs to scan all the data to estimate all possible split points for information gain, which takes considerable time and memory. LightGBM uses the histogram decision tree algorithm to make the memory footprint smaller and improve the calculation speed. It also uses the GOSS algorithm to calculate only high gradient data, which further saves space and

time overhead. The EFB algorithm is also used to bundle features, which reduces the dimension and time complexity of the algorithm. LightGBM uses a leafwise algorithm with depth constraints. In addition, it supports efficient parallel computing and increases the cache hit rate.

## 2.3 GRU

Cho K et al. [20] proposed an improved structure of the gating mechanism in 2014 to better solve the long-term dependency problem, GRU (gate recurrent unit), which optimizes the gate function of LSTM, combining the forget gate and the input gate in one update in the door. The update gate contains both the neuron state and the hidden state, which can reduce the complexity of the network unit, reduce the number of parameters, and greatly shorten the training time of the model. We use GRU to forecast time series; in contrast, timeliness is more important to the whole system, so we choose the GRU structure that consumes less time and has the same prediction effect as LSTM. A schematic diagram of its gate control structure is shown in Fig 1.

For each unit in the sequence, we denote $\sigma$ as a sigmoid function. *Tanh* represents a hyperbolic tangent function. $x_t$ is the input at time $t$. It is the implicit state $h_{t-1}$ of the moment $t-1$, which contains the dependency information of each previous moment. $r_t$ represents the reset gate, and $z_t$ stands for the update gate. means that the calculation logic of the two gates of the update gate is to splice the input of the current moment and the hidden state of the previous moment, and the output is controlled between [0, 1] through the sigmoid function. The calculation logic of the two gates is to join the input of the current moment and the hidden state of the last moment and control the output between [0, 1] through the sigmoid function. The output is inhibited as it approaches 0 and activated as it approaches 1.

First, the reset gate and update gate formulas are Eq (12) and Eq (13).

$$r_t = \sigma(W_{ir}x_t + b_{ir} + W_{hr}h_{t-1} + b_{hr}) \tag{12}$$

$$z_t = \sigma(W_{iz}x_t + b_{iz} + W_{hz}h_{t-1} + b_{hz}) \tag{13}$$

Then, the reset gate is used to reset the information, and the data are scaled to the range of

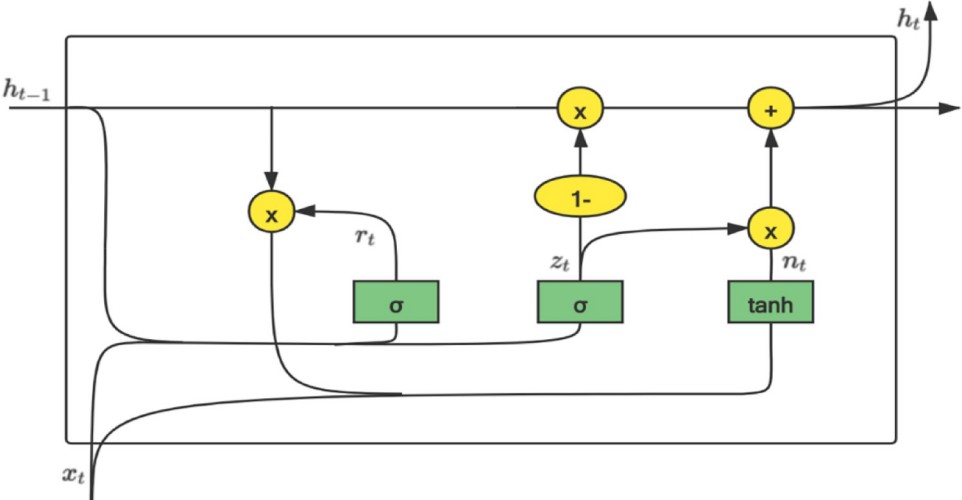

**Fig 1. Schematic diagram of the GRU structure.**

[−1, 1] by the *Tanh* function to obtain $n_t$. $n_t$ contains the information to be added at the current moment, which is equivalent to memorizing the state of the current moment, as given by Eq (14).

$$n_t = tanh(W_{in}x_t + b_{in} + r_t \odot (W_{hn}h_{t-1} + b_{hn})) \tag{14}$$

The last stage outputs the final hidden information. The function of this step is to forget some dimension information passed down and add some dimension information input by the current node. Output the current moment $y_t$ according to its hidden information, as given by Eq (15) and Eq (16).

$$h_t = (1 - z_t) \odot n_t + z_t \odot h_{t-1} \tag{15}$$

$$y^t = \sigma(W_0 h^t) \tag{16}$$

## 2.4 MMsAttention

Vaswani et al. [21] proposed a multihead attention mechanism, which uses different heads for different representation subspaces under the structure of the self-attentional mechanism. Multihead attention enables the model to jointly pay attention to different representational subspace information at different locations. In addition, multihead attention can also consider the information of different head positions to capture the intraday variation regularity of energy consumption more forcefully. In this paper, a multihead self-attention structure is proposed for the energy consumption prediction problem. As shown in Fig 2.

We take $x_i$ as input to the features corresponding to the decomposition of the volume of each hour of the day; there are 24 hours in a day, so the input of the masked multihead attention block is a vector sequence whose length M is 24. The sequence of vectors can be

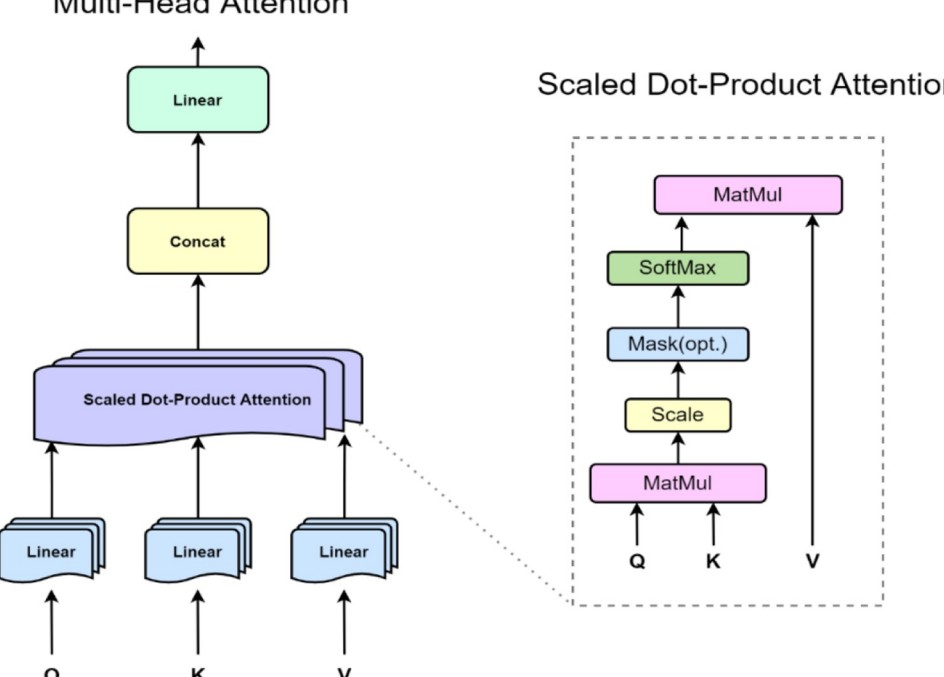

**Fig 2. Schematic diagram of the multihead self-attention structure.**

represented as an input matrix I, given by Eq (17).

$$I = x = (x_1, \ldots, x_M) \tag{17}$$

**2.4.1 Self-attention mechanism.** The essence of a self-attention function can be described as mapping a query and a set of key-value pairs to an output, where query, key, value, and output are all vectors. Each feature of the input has a set of vectors consisting of *q*, *k* and *v*. Q, K and V are the concatenation of all vector sequences of q, k and v, respectively. More intuitively, the attention mechanism is an operation that computes the similarity between a query and a key and extracts the query-related values for a weighted sum, as given by Eq (18), Eq (19), and Eq (20).

$$Q = W_q * I \tag{18}$$

$$K = W_k * I \tag{19}$$

$$V = W_v * I \tag{20}$$

Both $W_q$, $W_k$, $W_v$ are matrices that need to be updated through iterative training. The dimensions of the $Q$ and $K$ matrices corresponding to the input vector are both $d_k$, and the dimension of the $V$ matrix is $d_v$. In the attention matrix, the larger the value of the element is, the stronger the interaction relationship between energy consumption in different periods. The correlation matrix is given by Eq (21).

$$correlation_{matrix} = softmax\left(\frac{QK^T}{\sqrt{d_k}}\right) \tag{21}$$

The output of the self-attention layer is the weighted sum of the respective values, and the weight corresponding to each value is divided by the inner product of the corresponding query and key divided by the $\sqrt{d_k}$ of the corresponding key so that the inner product will not be too large. As is given by Eq (22).

$$Attention(Q, K, V) = Softmax\left(\frac{QK^T}{\sqrt{d_k}}\right) V \tag{22}$$

**2.4.2 Multi-head self-attention mechanism.** The multihead self-attention layer given by Eq (23) and Eq (24).

$$MultiHead(Q, K, V) = Concat[head_1, \ldots, head_h] W^O \tag{23}$$

$$head_i = Attention(QW_i^Q, KW_i^K, VW_i^V) \tag{24}$$

where the projections are parameter matrices $W_i^Q \in R^{d_{model} \times d_k}$, $W_i^K \in R^{d_{model} \times d_k}$, $W_i^V \in R^{d_{model} \times d_V}$, are head-specific weights for keys, queries and values, and $W_i^O \in R^{h \times d_V \times d_{model}}$ linearly combines outputs concatenated from all heads $head_i$.

The difference between the energy prediction and the text translation task is that the text translation can combine the following information of the input to output the above content translation. Energy consumption forecasting can only predict the future based on past energy consumption, so we propose a masked multihead attention structure. This masking ensures

that the predicted value at time $i$ can only rely on known outputs less than time $i$. The elements in $corraltion_{matrix}$ can be expressed as Eq (25).

$$C = [c_{ij}] \ where \ c_{ij} = 0, \forall i < j \tag{25}$$

$C$ is a lower triangular matrix, which ensures that the network will not see future information when making predictions and the results will not cheat us or make the prediction effect too accurate for us to rely on.

## 2.5 GRU-MMattention hybrid model architecture

As shown in Fig 3, to enable the network model to jointly pay attention to different representation subspace information at different locations, that is, to learn the interaction relationship between electricity consumption in different periods within a day, we added a masked-multi-head attention layer. Due to the large network depth, the Add&Norm layer is adopted to improve the prediction effect and speed up the network convergence. Without layer normalization, the gradient descent process is slow, and the descent trajectory fluctuates greatly. Then, the dimensionality is reduced with a feed-forward layer. Next, the data pass through the Add&Norm layer, and finally, the final result is output through the Linear layer.

## 2.6. GRU-MMattention-LightGBM model prediction process

Hybrid model training mainly includes three steps: feature construction, model training and prediction. The process is shown in Fig 4.

**Step 1, feature extraction on original data using Prophet.** After cleaning the original data, according to the Prophet decomposition method in section 2.1, the time series of energy consumption is decomposed into Trend, Seasonality and Residual. Then, features are built based on the data features as Eq (12).

**Step 2, model training.** The LightGBM model is trained on the training set. We pick the model with the best prediction performance on the validation set. After that, the parameters of the LightGBM model are frozen and saved. Then, the hyperparameters of GRU-MMattention are set and trained on the training set. The results obtained by the neural network and the results of LightGBM on the training set are used as the input of the multilayer perception (MLP) at the same time. The weights of both models are learned by the MLP. Finally, only the parameters of the neural network are iteratively adjusted according to the results of the validation set, and the best hybrid model is selected.

**Step 3, prediction.** The ensemble model trained in step 2 is saved, and prediction is performed on the test set.

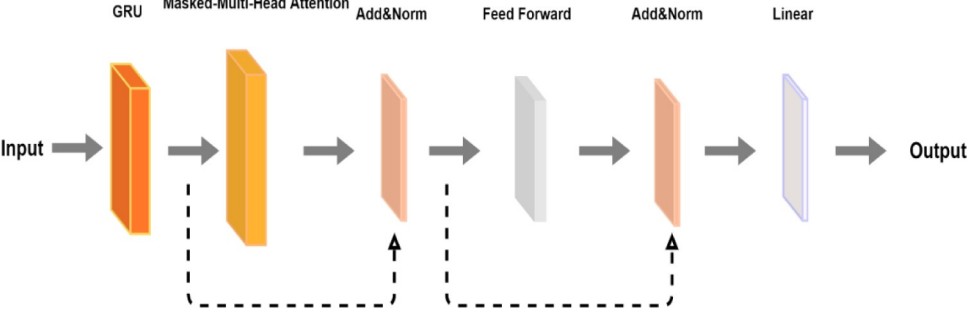

**Fig 3. GRU—MMattention structure attempts.**

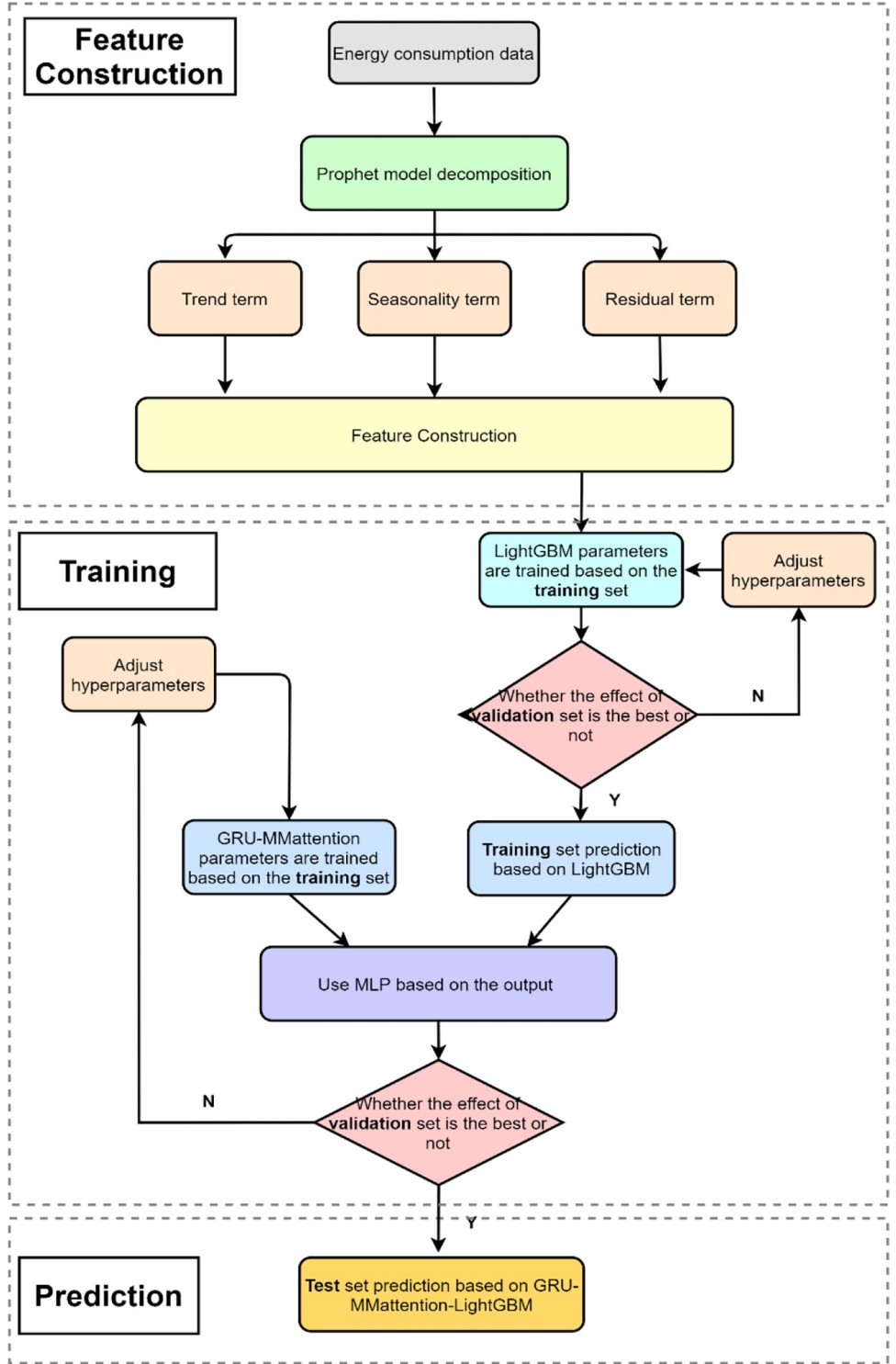

**Fig 4. GRU-MMattention-LightGBM model prediction process.**

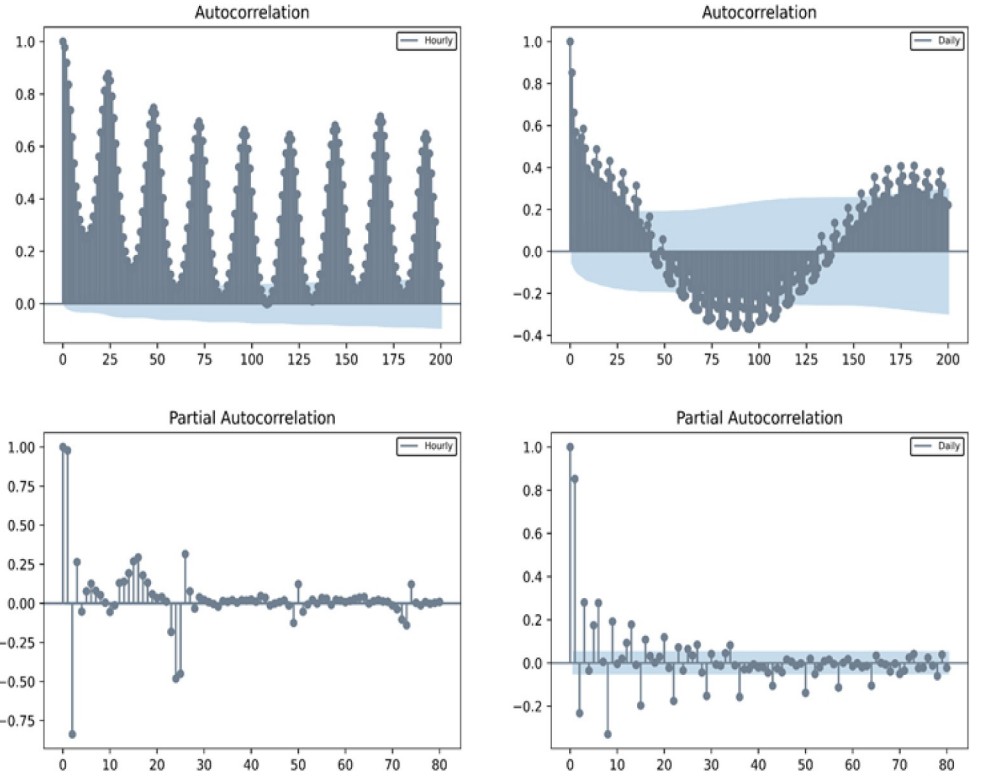

**Fig 5. ACF and PACF of electricity consumption per hour and daily.**

## 3. Empirical analysis

### 3.1 Data processing process

We selected the energy consumption data of the US PJM regional energy supply company in 14 regions from the Kaggle data website for research (https://www.kaggle.com/datasets/robikscube/hourly-energy-consumption). The energy consumption data from 0:00 on January 1, 2015, to 23:00 on August 3, 2018, were selected, with a frequency of 1 hour, and a total of 31440 samples. We performed k-neighbor imputation for the 4 missing values and removed 2 duplicate values. The first 85% of the data are the training set, 5% of the data are the validation set, and 10% of the data are the test set.

### 3.2 Feature window length selection

Fig 5 shows that the data of daily electricity consumption with the frequency of hours have a significant intraday cycle. The data of electricity consumption with a frequency of days have a significant seasonal effect. Combined with the censoring feature of PACF, the selection of the hyperparameter for the length of the feature window $K$ is 3. In addition, after the preexperiment of grid search with $K = [1, 6]$ when $K = 3$ on LightGBM and GRU, the test set performs best. Therefore, the window length $K$ of feature selection is 3.

### 3.3 Feature selection and normalization

Based on the cleaned data, Prophet decomposition is performed on the original data according to the method in section 2.1, as shown in Fig 6.

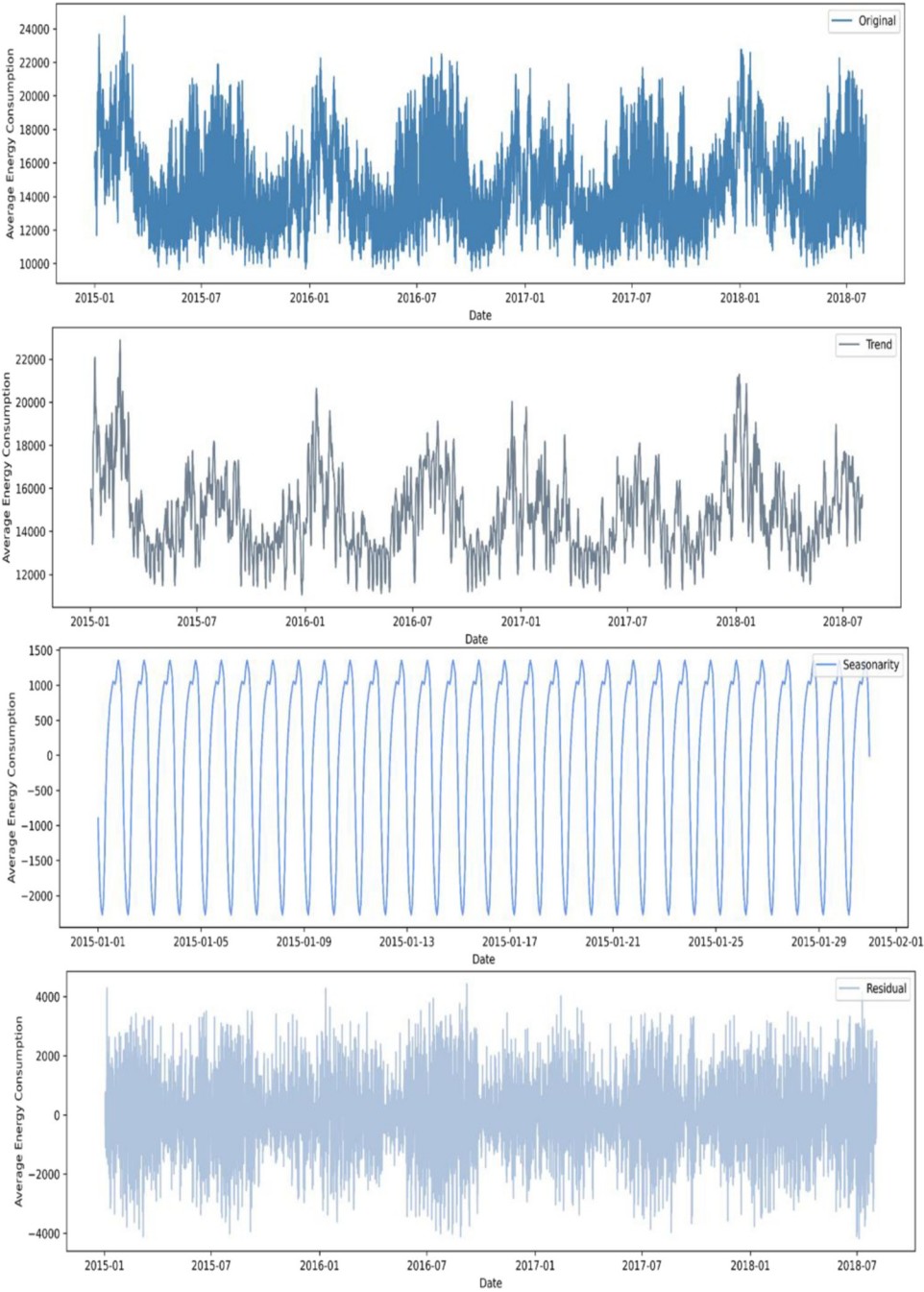

**Fig 6. Each feature after Prophet decomposition.**

It can be seen from subgraphs 1 and 2 of Fig 6 that the energy consumption cycle is decomposed very neatly, and the feature extraction is very effective.

To improve the convergence speed of the neural network, when the neural network model is established, minimum-maximum value normalization is performed on the features of each dimension in all samples so that the original data are in the range of [0, 1]. The normalized

equation is Eq (26).

$$x_{ij}^* = \frac{x_{ij} - min_{1 \leq i \leq N} x_{ij}}{max_{1 \leq i \leq N} x_{ij} - min_{1 \leq i \leq N} x_{ij}} \tag{26}$$

where $x_{ij}^*$ represents the normalized data, $x_{ij}$ represents the original data, N is the number of samples, $min_{1 \leq i \leq N} x_{ij}$ is the minimum value in the same dimension feature in all samples, and $max_{1 \leq i \leq N} x_{ij}$ is the maximum value in the same dimension feature in all samples.

Since the normalized data prediction is not the real predicted value, it is necessary to save the conversion factor to facilitate denormalization after the prediction and obtain the actual predicted value. The denormalization method is shown in Eq (27).

$$\hat{y} = (y_{max} - y_{min})\hat{y} + y_{min} \tag{27}$$

$\hat{y}$ represents the final predicted value after denormalization, $\hat{y}'$ represents the model prediction value under normalized data training, $y_{max}$ represents the maximum value of the labels in the test training set, and $y_{min}$ represents the minimum value of the labels in the test training set.

## 3.4 Evaluation indicators

This article discusses the problem of time series forecasting. Labels are numerical data, and we are more concerned with the gap between the actual value and the predicted value. Therefore, MAPE, MAE, MSE and RMSE are selected to measure the prediction accuracy and generalization ability of different models. $y_i$ represents the true value, $i = 1,2,...,N$. Eq (28), Eq (29), and Eq (30) are as follows:

$$MAPE = \frac{100\%}{N} \sum_{i=1}^{n} |\frac{\hat{y}_i - y_i}{y_i}| \tag{28}$$

$$MAE = \frac{1}{N} \sum_{i=1}^{n} |\hat{y}_i - y_i| \tag{29}$$

$$RMSE = \sqrt{\frac{1}{N} \sum_{i=1}^{n} (\hat{y}_i - y_i)^2} \tag{30}$$

## 3.5 Results

**3.5.1 GRU.** We used a grid search to adjust the hyperparameters in the preexperiment, and the hidden size ranged from {16,32,64,128}. The number of layers is {1,2,3,4}, and the group with the lowest MAPE and MAE is selected for comparison, as shown in Table 1.

**3.5.2 GRU- MMattention.** Due to the logical requirements of time series prediction, we use the masked multihead attention architecture, and the number of heads is set to 2. To capture the interaction between different periods within a day, we also added positional encoding to the architecture. The same GRU-MMattention also uses grid search to adjust the hyperparameters in the preexperiment, and the hidden size ranges from {16,32,64,128}. The number of

**Table 1. Comparison of the results of P-feature and TF-feature in GRU.**

|         | MAPE   | MAE     | RMSE     |
|---------|--------|---------|----------|
| P-GRU   | 10.35% | 1594.04 | 2011.069 |
| TF-GRU  | 12.15% | 1760.86 | 2230.38  |

**Table 2. Neural network hyperparameters.**

| Parameter | GRU setting | GRU-MMatten Setting |
|---|---|---|
| Activation Function | ReLU | ReLU |
| Optimization algorithm | Adam | Adam |
| Learning rate | 0.001 | 0.001 |
| Hidden size | 16 | 16 |
| Layer | 1 | 2 |
| Batch size | 64 | 64 |
| Epochs | 400 | 400 |

layers is {1, 2, 3, 4}. We select a set of hyperparameters with the lowest MAPE and MAE, as shown in Table 2. The prediction results of GRU and GRU-MMatten are shown in Fig 7.

For all samples predicted by this model, MAPE is 4.71%. The MAE is 700.38. The RMSE is 800.39.

**3.5.3 Light GBM.** Jiang X et al. [22] denoted LightGBM using moving temporal window features as TFLightGBM. We denote the LightGBM model of Prophet decomposition features as PlightGBM. Set the number of leaves as 31, the maximum depth as 5, the learning rate as 0.01, the number of estimators as 100, the minimum subtree weight as 0.01, and the minimum subtree sample as 20.

As Table 3 shows, PlightGBM is much more accurate than TFLightGBM, which shows that Prophet decomposition is significant in feature construction. The PlightGBM prediction results are shown in Fig 8.

**3.5.4 GRU-MMatten-LGB.** Based on the trained LightGBM model, including Prophet feature decomposition in section 3.5.3, the parameters of the LightGBM model are frozen and saved. Then, we set the hyperparameters of the optimized GRU-MMattention as in section 3.5.2, training on the training set. The results obtained by the neural network and the results of LightGBM on the training set are used as the input of the MLP simultaneously. The weights of both models are learned by the MLP. Set epochs = 400 when retraining the mixed model. The prediction results are shown in Fig 9.

For all samples predicted by the model, MAPE = 1.69%, MAE = 157.0741, and RMSE = 207.682.

(a)  (b)

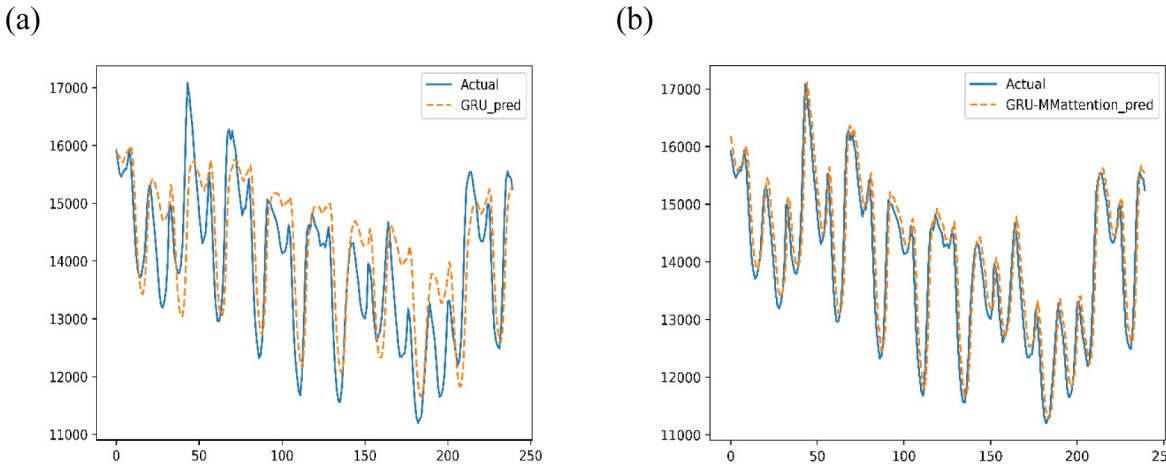

**Fig 7. The prediction results in the test set.** (a) GRU; (b) GRU-MMattention.

**Table 3. Comparison of the results of P-feature and TF-feature in LightGBM.**

|  | MAPE | MAE | RMSE |
|---|---|---|---|
| TFLightGBM | 4.6% | 657.14 | 700.76 |
| PlightGBM | 3.71% | 530.62 | 631.86 |

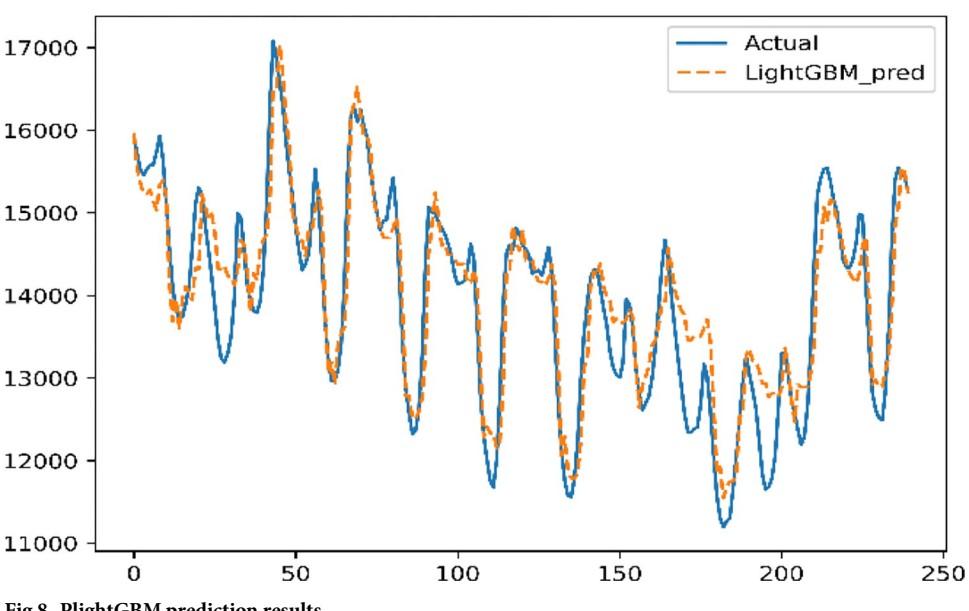

**Fig 8. PlightGBM prediction results.**

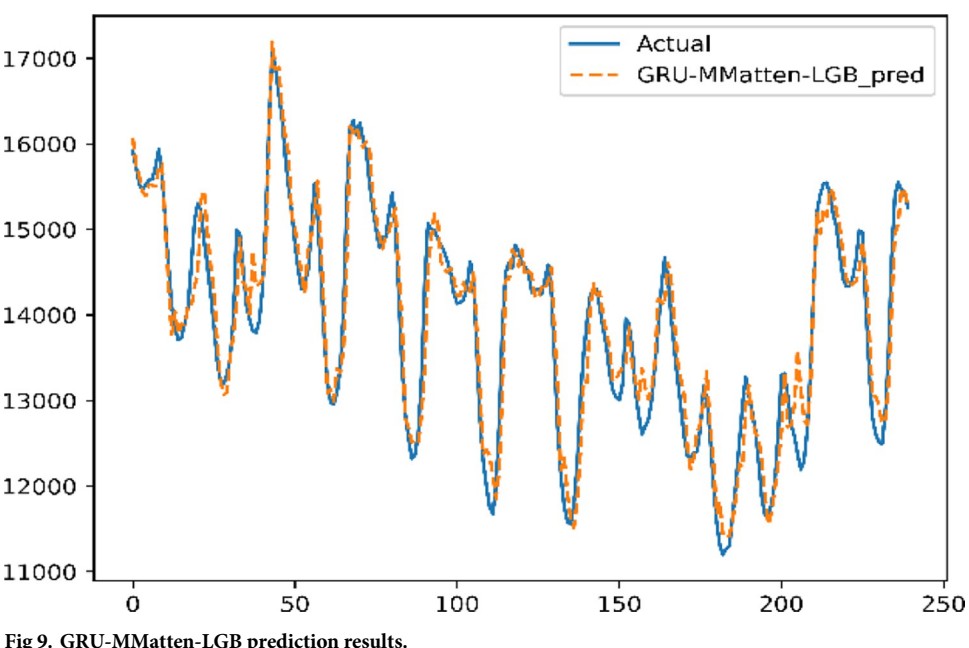

**Fig 9. GRU-MMatten-LGB prediction results.**

**Table 4. Comparison of all optimal model results.**

|  | GRU | LightGBM | GRU-MMattention | GRU-MMatten-LGB |
|---|---|---|---|---|
| MAPE | 10.35% | 3.71% | 4.71% | 1.69% |
| MAE | 1594.041 | 530.617519 | 700.3894 | 157.0741 |
| RMSE | 2011.069 | 631.860881 | 839.8034 | 207.682 |

## 3.6 Model comparison results

**3.6.1 Comparison of the methods' accuracy.**   Based on the results in 3.5, we compare the optimal models in each structure, as shown in Table 4 and Fig 10.

**3.6.2 Comparison of the antinoise ability of the methods.**   The strong anti-noise capability in this paper refers to the strong anti-interference capability of the model against the noise of the training set. Specifically, if noise appears in the training data set and the model learns the data contaminated by noise, the prediction result is still not much different from the data without noise; then, we can conclude that the model has a good anti-noise ability. We selected the energy consumption data of PJM regional power supply companies, different from those in the previous areas. The power consumption data from 0:00 on January 1, 2017 to 23:00 on August 3, 2018 were selected, with a frequency of 1 hour, and a total of 11380 samples. After data cleaning, we added noise to 10% of the data in the training set, Noise~N(0,1000), where 1000 is the standard deviation of the power consumption data itself. Finally, the model is retrained, and metrics are calculated on the test set. The experiment was repeated 10 times, and the metrics were recorded each time. After the t test of the 10 experimental results, if the metrics did not change significantly, the GRU-MMatten-LGB method could be considered to have a strong anti-noise ability.

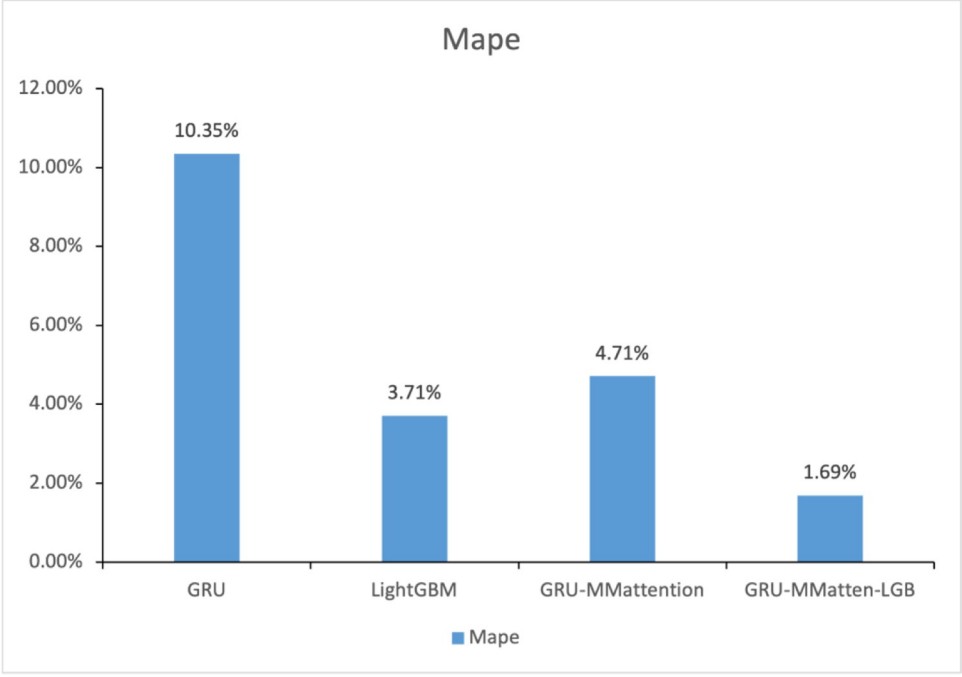

**Fig 10. Comparison of all model results.**

Table 5 shows that, without noise, the MAPE of GRU is 0.028152. Mape for LightGBM is 0.026251. Mape for GRU-attention is 0.020580. GRU-MMatten-LGB's MAPE is 0.019802.

$$H_0 : Distribution\ of\ MAPE\ on\ noise\ experient \sim N(u, \sigma^2)$$

$$H_1 : Distribution\ of\ MAPE\ on\ noise\ experient\ doesn't\ follow\ N(u, \sigma^2)$$

Table 7 shows that the above p values are all greater than 0.1, so there is no reason to reject the null hypothesis. Therefore, we conclude that the distribution of MAPE in the noise experiment conforms to a normal distribution. Then, we can check whether the MAPE mean of the noise experiment is smaller than the given population mean, which is denoted as $MAPE_{normal}$

$$H_0 : Mean_{noise} \leq MAPE_{normal}$$

$$H_1 : Mean_{noise} > MAPE_{normal}$$

$Mean_{noise}$ represents the mean of the MAPE on a certain model in 10 noise experiments, and the data are shown in Table 6.

$Mean_{normal}$ represents the MAPE of a certain model without noise, and the data are shown in Table 5. The results of the t test are shown in Table 8.

**Table 5. MAPE of all methods without noise.**

|  | GRU | LightGBM | GRU-attention | GRU-MMatten-LGB |
|---|---|---|---|---|
| MAPE | 0.028152 | 0.026851 | 0.023380 | 0.019802 |

**Table 6. The MAPE of the noise experiment on all methods.**

| GRU | LightGBM | GRU-MMattention | GRU-MMatten-LGB |
|---|---|---|---|
| 0.032619 | 0.026836 | 0.022095 | 0.019150 |
| 0.034004 | 0.027281 | 0.024159 | 0.020839 |
| 0.031138 | 0.025671 | 0.02111 | 0.019386 |
| 0.032464 | 0.028517 | 0.023074 | 0.020019 |
| 0.033966 | 0.027891 | 0.022989 | 0.021202 |
| 0.035404 | 0.026632 | 0.025145 | 0.020101 |
| 0.035194 | 0.02885 | 0.023803 | 0.019281 |
| 0.036681 | 0.026585 | 0.026043 | 0.019147 |
| 0.027001 | 0.02813 | 0.023191 | 0.019386 |
| 0.034243 | 0.025368 | 0.02813 | 0.019852 |

**Table 7. The results of the Shapiro–Wilk test.**

|  | GRU | LightGBM | GRU-MMattention | GRU-MMatten-LGB |
|---|---|---|---|---|
| Statistic | 0.89618 | 0.92029 | 0.94990 | 0.92778 |
| p-vlaue | 0.16597 | 0.32113 | 0.64277 | 0.38887 |

**Table 8. The results of the t test.**

|  | GRU | LightGBM | GRU-MMattention | GRU-MMatten-LGB |
|---|---|---|---|---|
| Statistic | 6.57063 | 2.52219 | 5.84248 | 0.19713 |
| p-vlaue | 3.153355e-05 | 0.01513 | 8.16317e-05 | 0.42405 |

According to the results of Tables 6 and 8, the p value of the t test in the GRU noise experiment is less than 0.1. The mean MAPE of the GRU model in 10 noise experiments is 0.03327, which can be considered to be significantly higher than 0.028152 when noise is not added. Therefore, we can conclude that the GRU model has poor anti-noise ability. Similarly, the p value of the t test in the GRU-MMattention noise experiment is less than 0.1, and we can also draw the conclusion that GRU-MMattention has poor anti-noise ability.

However, for the LightGBM method in the noise experiment, its p value of the t test is more than 0.1. The mean MAPE on LightGBM in 10 noise experiments is 0.02720, which cannot be judged to be significantly greater than 0.026851 without noise. It can be considered that the impact of noise on the prediction results of the LightGBM model is not significant, and the LightGBM method has a strong anti-noise ability. Similarly, the p value of the t test on GRU-MMatten-LGB is also more than 0.1. It can be concluded that GRU-MMatten-LGB has a strong anti-noise ability.

## 4. Conclusion

Based on PJM District Energy Company's energy consumption data in 14 districts from January 1, 2015, to August 3, 2018, our conclusions are as follows:

1. For feature construction, the method based on prophet decomposition is more predictive than the simple time window method. The MAPE, MAE and MSE of PlightGBM were 3.71%, 530.62 and 631.86, respectively. The MAPE, MAE and MSE of TFLightGBM are 4.6%, 657.14 and 700.76, respectively. The prediction accuracy of LightGBM using Prophet features is improved by 1.1%. Similarly, P-GRU improves the prediction accuracy by approximately 2% over TFGRU. Therefore, Prophet decomposition is effective for energy consumption prediction and can be regarded as a good distributed representation in representation learning.

2. As far as a single model is concerned, GRU-MMattention can improve GRU, and the masked multihead-attention architecture has reference significance for energy consumption prediction. The MAPE, MAE and MSE of P-GRU were 10.35%, 1594.04 and 2011.069, respectively. For the GRU-MMattention architecture, MAPE = 4.71%, MAE = 700.38, and RMSE = 800.39. The GRU-MMattention architecture is approximately 6% better than GRU. The combination of the gating-attention mechanism is more stable and accurate than a single gating mechanism.

3. Comparing the evaluation indicators of the four model architectures of GRU, GRU-MMattention, LightGBM, and GRU-MMatten-LGB, it can be clearly seen that the structure of GRU-MMatten-LGB is practical and effective for energy prediction. The MAPE of the GRU-MMattention-LightGBM model is 1.69%, which is 8.66% lower than the GRU structure relative error and 2.02% lower than the LightGBM prediction relative error. Compared with the single model, the prediction accuracy and stability of the hybrid architecture have been significantly improved.

4. Although GRU and GRU-MMattention are poor in anti-noise ability, the GRU-MMatten-LGB structure has the characteristics of a strong anti-noise ability of the LightGBM method, which can reduce the impact of the energy consumption mutation point on the overall prediction effect of the model.

## Supporting information

**S1 Data.**
(ZIP)

## Author Contributions

**Conceptualization:** Shaokun Liang, Ningxian Liu, Xuchu Jiang.

**Data curation:** Shaokun Liang, Tao Deng, Anna Huang, Ningxian Liu, Xuchu Jiang.

**Formal analysis:** Tao Deng, Anna Huang, Ningxian Liu.

**Investigation:** Anna Huang.

**Methodology:** Ningxian Liu.

**Project administration:** Ningxian Liu.

**Validation:** Ningxian Liu.

**Visualization:** Ningxian Liu.

**Writing – review & editing:** Ningxian Liu.

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
