## [Decision Letter · Decision Letter 0]

28 Apr 2022

PONE-D-22-02103Energy consumption prediction using GRU-MMattention-LightGBM model with features of Prophet decompositionPLOS ONE

Dear Dr. Jiang,

Thank you for submitting your manuscript to PLOS ONE. After careful consideration, we feel that it has merit but does not fully meet PLOS ONE’s publication criteria as it currently stands. Therefore, we invite you to submit a revised version of the manuscript that addresses the points raised during the review process.

We look forward to receiving your revised manuscript.

Kind regards,

**Le Hoang Son, Ph.D**

Academic Editor

PLOS ONE

Journal Requirements: 

**Comments to the Author**

1. Is the manuscript technically sound, and do the data support the conclusions?

Reviewer #1: Partly

2. Has the statistical analysis been performed appropriately and rigorously? 

Reviewer #1: Yes

3. Have the authors made all data underlying the findings in their manuscript fully available?

Reviewer #1: Yes

4. Is the manuscript presented in an intelligible fashion and written in standard English?

Reviewer #1: No

5. Review Comments to the Author

**Reviewer #1**: 

The paper proposes a hybrid machine learning model, GRU-MMattention-LightGBM, with feature selection based on prophet decomposition.

1. The contributions are not presented clearly in Section 1. After the literature survey, the authors should explain technology gaps and the paper's novelty.

2. The authors stated that the proposed method has a strong anti-noise ability, which can reduce the impact of mutation points. This capability needs to be verified and discussed in the case study.

3. The font in Figures 5 and 6 is too small to read.

4. There are so many typos and grammatical errors as follows:

Line 53: Jianet al.

Line 85: multiple spaces

Line 88: T The

Line: 312: multiple spaces

5. Usually, the conjunction "And" does not come as the first sentence word.

---

## [Author Response · Author response to Decision Letter 0]

16 May 2022

Dear Editors and Reviewers:

Thank you for your letter and for the reviewer's comments concerning our manuscript entitled “Energy consumption prediction using the GRU-MMattention-LightGBM model with features of Prophet decomposition” (PLOS ONE). These comments are all valuable and very helpful for revising and improving our paper and have important guiding significance to our research. We have studied the comments carefully and have made corrections that we hope will meet with approval. The main corrections in the paper and the responses to the reviewer’s comments are as follows.

Reviewer #1:

The authors well have handled the reviewer's comment. I have additional comments for this revision as follows;

Comment 1:

1. The contributions are not presented clearly in Section 1. After the literature survey, the authors should explain technology gaps and the paper's novelty. 

Thank you for your comments. 

We reorganized the literature, pointed out the shortcomings of current techniques and supplemented the novelty of our proposed methods. 

New content is added on line 69：

However, RNN-based models are sensitive to different random weight initializations and thus prone to overfitting. Therefore, the single network structure lacks predictive stability and generalization ability.

New content is added on line 79：

Without sufficient subsampling，randomized trees can lead to overfitting, according to Speiser J et al.[22]

New content is added on line 83：

However, their deep neural network model is also sensitive to different random weight initializations, which is very unstable and does not combine with machine learning methods. 

New content is added on line 86：

However, GRU is used for feature extraction in their work, the periodicity of energy consumption data is not well captured, and GRU-attention has higher prediction accuracy than GRU. 

New content is added on line 92：

This method only uses an attention structure, and has a poor ability to learn the periodic characteristics of power consumption, and the accuracy is not as good as that of GRU-attention. 

New content is added on line 95：

LightGBM, a single machine learning model, has strong predictive ability in general time series, but its accuracy in predicting periodic time series is still inadequate. Deep learning methods, such as LSTM, GRU and other RNN-based models, rely heavily on initialization parameter selection, and the model stability and accuracy is not good enough. A single GRU or a single attention is not as good at predicting periodic time series as a combination of the two. Current research cannot solve the above three key problems simultaneously. 

Comment 2:

2. The authors stated that the proposed method has a strong anti-noise ability, which can reduce the impact of mutation points. This capability needs to be verified and discussed in the case study.

Response:

Thank you for your comments.

Noise error refers to the difference between the label value of the data and the real value, which belongs to the measurement error. The strong anti-noise capability in this paper refers to the strong anti-interference capability of the model against the noise of the training set. Specifically, if noise appears in the training data set and the model learns the data contaminated by noise, the prediction result is still not much different from the data without noise, then we can conclude that the model has a good anti-noise ability. Based on the above ideas, we supplement a new noise experiment in Section 3.6.

We selected the energy consumption data of PJM regional power supply company, different from those in the previous areas. The power consumption data from 0:00 on January 1, 2017 to 23:00 on August 3, 2018 were selected, with a frequency of 1 hour, and a total of 11380 samples. After data cleaning, we added Noise to 10% of the data in the training set, Noise~N(0,1000), where 1000 is the standard deviation of the power consumption data itself. Finally, the model is retrained and metrics are calculated on the test set. Repeat the experiment 10 times and record the metrics for each time. After the t-test of the 10 experimental results, if metrics did not change significantly, the GRU-MMatten-LGB method could be considered to have a strong anti-noise ability.

Table 6 shows the MAPE of 4 different models on the test set after randomly adding normal distribution noise to 10% of the training set. Since the premise of t test is that the population of the MAPE in the nosie experiment follows normal distribution, we first adopt Shapiro-Wilk test on the 10 results of each model to judge whether they follow normal distribution.

H_0:Distribution of MAPE on noise experient ~N(u,σ^2) 

H_1:Distribution of MAPE on noise experient doesn^' t follow N(u,σ^2) 

It can be seen from Table 7 that the above p-values are all greater than 0.1, so there is no reason to reject the null hypothesis. Therefore, we conclude that the distribution of MAPE in noise experiment conforms to normal distribution. Then we can check whether the MAPE mean of the noise experiment is smaller than the given population mean, which is denoted as MAPE_normal

H_0:Mean_noise≤MAPE_normal

H_1:Mean_noise>MAPE_normal

Mean_noise represents the mean of the MAPE on a certain model in 10 noise experiments, and the data are shown in Table 6.

Mean_normal represents the MAPE of a certain model without noise, and the data are shown in Table 5. The results of the t-test would be shown in Table 8.

According to the results of Table 6 and Table 8, p-vlaue of t-test in GRU noise experiment is less than 0.1. The mean of MAPE on GRU model in 10 noise experiments is 0.03327, which can be considered to be significantly higher than 0.028152 when noise is not added. Therefore, we can conclude that GRU model has poor anti-noise ability. Similarly, the p-value of the t-test in GRU-MMattention noise experiment is less than 0.1, we can also draw the conclusion that GRU-MMattention is poor in anti-noise ability.

However, LightGBM method in the noise experiment, its p-vlaue of t-test is more than 0.1. The mean of MAPE on LightGBM in 10 noise experiments is 0.02720, which cannot be judged to be significantly greater than 0.026851 without noise. It can be considered that the impact of noise on the prediction results of LightGBM model is not significant, and the LightGBM method has a strong anti-noise ability. Similarly, the p-vlaue of t-test on GRU-MMatten-LGB is also more than 0.1.It can be concluded that GRU-MMatten-LGB has a strong anti-noise ability.

Comment 3:

3. The font in Figures 5 and 6 is too small to read.

Response:

Thank you for your comments. The Figures 5 and 6 have been revised as follows.

Comment 4:

4. There are so many typos and grammatical errors as follows:

Line 53: Jianet al.

Line 85: multiple spaces

Line 88: T The

Line: 312: multiple spaces

Response:

Thank you for your comments. We are very sorry for our careless mistake and it was rectified at Line 53, 85, 88, 312. At the same time, we checked the full text.

Comment 5:

5. Usually, the conjunction "And" does not come as the first sentence word.

Response:

We appreciate it very much for this good suggestion, and we have done it according to your ideas. We revised the abstract and Section 2.2 with this problem.

Responses to the Editor:

The modified manuscript has been improved and polished by professional services. All modifications have been marked in RED, and a point-by-point reply has been completed. We tried our best to improve the manuscript and made some changes in the manuscript. Thank you very much for your thorough review and valuable advice. Your comments and suggestions have made a great contribution to the improvement of this paper's quality.

Thank you and best regards.

Xuchu Jiang

xuchujiang@zuel.edu.cn

School of Statistics and Mathematics

Zhongnan University of Economics and Law

---

## [Decision Letter · Decision Letter 1]

20 Oct 2022

Energy consumption prediction using the GRU-MMattention-LightGBM model with features of Prophet decomposition

PONE-D-22-02103R1

Dear Dr. Jiang,

We’re pleased to inform you that your manuscript has been judged scientifically suitable for publication and will be formally accepted for publication once it meets all outstanding technical requirements.

Kind regards,

Shuo-Yan Chou

Academic Editor

PLOS ONE

Additional Editor Comments (optional):

Reviewers' comments:

Reviewer's Responses to Questions

**Comments to the Author**

1. If the authors have adequately addressed your comments raised in a previous round of review and you feel that this manuscript is now acceptable for publication, you may indicate that here to bypass the “Comments to the Author” section, enter your conflict of interest statement in the “Confidential to Editor” section, and submit your "Accept" recommendation.

Reviewer #1: All comments have been addressed

2. Is the manuscript technically sound, and do the data support the conclusions?

Reviewer #1: Yes

3. Has the statistical analysis been performed appropriately and rigorously? 

Reviewer #1: Yes

4. Have the authors made all data underlying the findings in their manuscript fully available?

Reviewer #1: Yes

5. Is the manuscript presented in an intelligible fashion and written in standard English?

Reviewer #1: Yes

6. Review Comments to the Author

Reviewer #1: The authors have adequately addressed all the comments raised by this reviewer. No more comments.

7. PLOS authors have the option to publish the peer review history of their article (what does this mean?). If published, this will include your full peer review and any attached files.

Reviewer #1: No

---

## [Editor Report · Acceptance letter]

21 Nov 2022

PONE-D-22-02103R1 

Energy consumption prediction using the GRU-MMattention-LightGBM model with features of Prophet decomposition 

Dear Dr. Jiang:

I'm pleased to inform you that your manuscript has been deemed suitable for publication in PLOS ONE. Congratulations! Your manuscript is now with our production department. 

Kind regards, 

on behalf of

Professor Shuo-Yan Chou 

Academic Editor

PLOS ONE